# An Embedding-Based Semantic Analysis Approach: A Preliminary Study on Redundancy Detection in Psychological Concepts Operationalized by Scales

**DOI:** 10.3390/jintelligence13010011

**Published:** 2025-01-16

**Authors:** Zhen Huang, Yitian Long, Kaiping Peng, Song Tong

**Affiliations:** 1School of Social Sciences, Tsinghua University, Beijing 100084, China; huang_zhen@tsinghua.edu.cn (Z.H.); pengkp@tsinghua.edu.cn (K.P.); 2Wuhan Britain-China School, Wuhan 430000, China; kylelongyitian@gmail.com; 3Department of Psychological and Cognitive Sciences, Tsinghua University, Beijing 100084, China

**Keywords:** redundancy detection, psychological scales, GPT, hierarchical clustering

## Abstract

As psychological research progresses, the issue of concept overlap becomes increasing evident, adding to participant burden and complicating data interpretation. This study introduces an Embedding-based Semantic Analysis Approach (ESAA) for detecting redundancy in psychological concepts, which are operationalized through their respective scales, using natural language processing techniques. The ESAA utilizes OpenAI’s text-embedding-3-large model to generate high-dimensional semantic vectors (i.e., embeddings) of scale items and applies hierarchical clustering to group semantically similar items, revealing potential redundancy. Three preliminary experiments evaluated the ESAA’s ability to (1) identify semantically similar items, (2) differentiate semantically distinct items, and (3) uncover overlap between scales of concepts known for redundancy issues. Additionally, comparative analyses assessed the ESAA’s robustness and incremental validity against the advanced chatbots based on GPT-4. The results demonstrated that the ESAA consistently produced stable outcomes and outperformed all evaluated chatbots. As an objective approach for analyzing relationships between concepts operationalized as scales, the ESAA holds promise for advancing research on theory refinement and scale optimization.

## 1. Introduction

As psychological research progresses, an increasing number of concepts and scales have been introduced, yet limited attention has been given to the potential overlaps among them. These overlaps could increase participant burden ([14]), potentially compromising data quality and complicating the interpretation of research findings ([18]; [10]; [11]). More critically, redundancy hinders the field from advancing toward theoretical simplicity and precise measurement. Consequently, addressing redundancy in psychological concepts and their associated scales has become an urgent priority ([20]; [4]; [37]).

Traditionally, redundancy in psychological scales has been identified through methods such as factor analysis, correlational studies, and expert judgment ([19]; [9]; [15]). While these methods offer valuable insights, they also present limitations. Factor analysis and correlational studies depend on empirical data, which can be costly and time-consuming to gather, with the added risk of survey errors ([12]; [36]; [33]). Expert judgment, while insightful, is inherently subjective, making it difficult to ensure the reproducibility of research findings ([2]; [26]; [39]).

In recent years, advancements in natural language processing (NLP) and machine learning have created new opportunities for text data analysis. Notably, transformer-based language models, such as Bidirectional Encoder Representations from Transformers (BERT), Generative Pre-trained Transformer (GPT) have demonstrated remarkable capabilities in understanding and representing semantic content ([41]; [16]; [7]). These large language models (LLMs) have found increasing applications in educational and psychometric research, especially for analyzing psychological scales and identifying redundancies (e.g., [5]; [27]; [40]). At the core of these approaches is the use of text embeddings, which map words, phrases, or sentences into high-dimensional vectors to capture semantic relationships ([6]; [29]; [31]; [32]). This has enabled researchers to assess conceptual overlaps by comparing the semantic structures of scale items, revealing redundancies that align with theoretical models of non-overlapping concepts ([21]; [34]; [3]; [42]).

One of the earliest applications of embedding-like techniques in this domain was the Semantic Scale Network ([34]), which used Latent Semantic Analysis (LSA) to quantify semantic overlaps between psychological scales. LSA relied on vectorizing textual data into a latent topic space and measuring semantic distances to identify conceptual similarities. This allowed researchers to analyze scales without relying on participant data, marking a significant step forward. However, LSA’s reliance on static, predefined datasets (4023 scales) limited its ability to capture semantic representations from the co-occurrence. [23] ([23]) addressed these limitations by adopting BERT embeddings, which provided richer semantic representations. They further fine-tuned BERT using a domain-specific database (SBERT) to enhance its performance in psychological survey tasks. Another key contribution of their work was providing extensive validation of their approach against empirical psychometric data, while they did not evaluate the model in these data.

Building on these efforts, we introduce a technique solution to the redundancy detection of psychological concepts and scales that does not require the involvement of human participants: the Embedding-based Semantic Analysis Approach (ESAA). The ESAA leverages the power of advanced language models to generate high-dimensional vector representations (embeddings) of scale items. Then, it applies unsupervised clustering algorithms and other techniques to perform semantic analysis on these embeddings, providing a more objective and efficient means of detecting redundancy compared to traditional methods. Notably, some ongoing studies are also exploring the use of embeddings to address the concept redundancy problem ([23]; [42]). For example, [42] ([42]) transitioned to MPNet, a lightweight transformer-based LLM, which showed superior performance in semantic parsing compared to fine-tuned BERT. They introduced a data-driven approach to address clustering challenges, testing multiple clustering numbers to optimize their solutions for empirical data. While this approach is reported to improve performance, its implementation required significant computational resources, and its validation remains not completely convincing for being heavily dependent on empirical datasets. Moreover, the reliance on static embeddings, whether fine-tuned BERT or MPNet, remains a limitation in rapidly evolving research contexts. The inherent superiority of LLMs in semantic representation raises questions about whether incremental optimizations, such as clustering number selection, justify the effort when the baseline performance of LLMs is already strong. As LLMs continue to evolve, the value of such resource-intensive refinements diminishes, potentially rendering these models and their associated optimizations obsolete.

Our study adopts a distinct roadmap compared to contemporaneous research. The ESAA follows a multi-step process, starting with transforming scale items into high-dimensional vector representation using OpenAI’s text-embedding-3-large model (referred as GPT-3 large model hereafter). Then, the ESAA applies unsupervised clustering algorithms and related computational techniques to conduct semantic analysis on these embeddings. The main output of the ESAA is a systematic framework referred to as the ESAA-generated framework (EGF) that delineates the semantic relationships among items, accompanied by their associated synthetic correlations.

As the initial phase of our research, we conducted a fundamental evaluation of the ESAA’s performance against essential baseline requirements. Instead of using empirical data, we adopted theoretical predictions from established psychological frameworks as evaluation criteria. This methodological choice was made to ensure objective and well-defined assessment standards, minimizing potential biases and measurement errors inherent in empirical benchmarks.

Thus, the validation strategy involves selecting several well-established psychological constructs with clear theoretical relationships, then applying the ESAA to output its analysis on the material, and finally comparing the ESAA’s output against theoretical predictions. The ESAA will be considered for further investigation for more rigorous validation standard only if it meets the baseline requirements in the current study, otherwise, further exploration should not be warranted.

Following this strategy, our initial experiments evaluated three key capabilities of the ESAA: (1) the convergence of semantically similar items, (2) the discrimination between items with significant semantic differences, and (3) the identification of overlapping patterns among concepts known for redundancy issues. Additionally, we conducted a robustness check and comparative analyses to assess the incremental value of the ESAA over that of state-of-the-art chatbots, such as ChatGPT-4. The findings indicate that the ESAA’s performance meets the fundamental requirements for the specific experimental materials tested, supporting its progression to more comprehensive testing in future research.

This paper is structured as follows: First, we introduce the principles and architecture of the ESAA. Next, we describe a series of preliminary validation experiments conducted to evaluate the ESAA’s performance. Subsequently, we discuss the key insights derived from these experiments, compare our research with related studies, and highlight the limitations of the current research. Finally, we outline directions for future work to further enhance and validate the ESAA.

## 2. The Proposed Approach: The ESAA

The ESAA is designed to help psychologists investigate conceptual redundancy by identifying overlaps among constructs operationalized as their scales. The ESAA follows a clear multi-step process, as shown in Figure 1 (overview) and Figure 2 (detailed conceptual diagrams). First, the ESAA uses OpenAI’s text-embedding-3-large model (referred to as GPT-3 large model hereafter) to generate semantic embeddings for scale items, converting textual data into high-dimensional vectors (see Figure 2A). For example, in Figure 2A, the orange bar represents the input material processed by the ESAA, taken from Experiment 1 in the following section, consisting of 12 text-based items from psychological scales. The green rectangle illustrates the embedding for an individual item as a row, while the purple rectangle represents the embedding’s 3072-dimensional structure generated by the GPT-3 large model. Next, the ESAA computes semantic distances between these embeddings to capture relationships among scale items. As illustrated in Figure 2B, semantic distance calculation uses mathematical concepts such as Euclidean distance and cosine similarity. Since visualizing a 3072-dimensional space directly is not practical, Figure 2B uses a 3D coordinate system to represent these concepts, where each data point corresponds to an embedding from Figure 2A. A pairwise cosine similarity matrix is also shown, summarizing the semantic relationships across all item embeddings. To replace traditional empirical correlations, the ESAA introduces a new metric, synthetic correlation, which assesses these relationships based entirely on semantic embeddings. Finally, unsupervised clustering techniques are applied to identify latent structures within the digitized items, providing insights into redundancies among psychological constructs. As shown in Figure 2C, the steps of agglomerative hierarchical clustering are outlined, illustrating how items are grouped based on their semantic similarities. The following sections explain the key steps of the ESAA in detail.

### 2.1. Step 1. Data Processing: Text Vectorization

The first key operation of the ESAA is to convert the semantics of text-based scale items into computable formats. This is achieved by transforming the text into numerical representations known as embeddings. In the context of large language models (LLMs), embeddings are high-dimensional vector representations that capture the semantic meaning of the text, enabling comparison across different texts. LLMs, such as GPT-3, are based on transformer architectures, which rely on self-attention mechanisms to process text. When a text input (e.g., a psychological scale item) is provided to an LLM, the model breaks it into smaller units (tokens). Each token is mapped to an initial embedding vector from a learned vocabulary. The transformer layers then refine these embeddings by iteratively updating them through attention mechanisms and feed-forward neural networks. During this process, each token’s embedding is adjusted based on its contextual relationship with other tokens in the input sequence. The final output embeddings encapsulate the individual word meanings and their semantic relationships within the given context. For applications like the ESAA, the model generates a single embedding for the entire text input (sentence or phrase), representing its overall semantic meaning.

Selecting an appropriate LLM is crucial in that embeddings are generated by these models, which rely on advanced neural network architectures. By September 2024, the final design phase of the ESAA, a wide range of LLMs are available, either as open-source resources or through Application Programming Interfaces (API) based services, with varying capabilities in representing the semantics of textual material. For example, BERT model, a widely used model with 768-dimensions embeddings and notable for its extensive use in academic research ([24]). On the other hand, LLMs with higher-dimensional embeddings have been reported to capture multi-faced semantic meanings ([38]), such as concept activation and user performance. Concretely, OpenAI’s text-embedding-3-small model produces embeddings with 1536 dimensions., and GPT-3-large model stands out as more powerful, offering embeddings with 3072 dimensions. As OpenAI’s latest released model, GPT-3 large model size and diverse pretraining enable it to capture richer semantic relationships than models like BERT, which is critical for redundancy detection in psychological scales (e.g., [7]). Additionally, our preliminary study showed that embeddings generated by the BERT model were significantly distant from validation (see Appendix A), while GPT-3 produced more cohesive clusters that aligned better with theoretical constructs. Therefore, we chose the GPT-3 large model as LLM for embedding generation in the ESAA. The embedding generation process is conceptually shown in Figure 1 and Figure 2 (Panel for Step 1).

The vectorization process involves sending the text input (i.e., psychological scale items after necessary preprocessing) to the API endpoint provided by OpenAI, which returns the corresponding embeddings. The API handles the tokenization and encoding processes internally, ensuring consistent and high-quality embedding vectors. The vectorizing were performed using Python 3.11.3 in this study, with the openai libraries employed. An example of the embedding for a scale item is provided in Appendix A.

### 2.2. Step 2. Algorithmic Computation: Semantic Distance and Structure Exploration

The second key step in the ESAA is the analysis of embeddings, divided into two sub-steps. First, we calculate the semantic distance or similarity between items in the semantic space (similarity and distance are mathematically related, which are used interchangeably in this study as two sides of the same coin). Second, we analyze the relationships among items based on these semantic distances, focusing on the internal structure of the item set as a whole.

#### 2.2.1. Semantic Distance

Semantic distance is an important metric in this study, reflecting the relationships between embeddings. A smaller semantic distance indicates a closer approximation of the semantics of the embeddings. Several methods are available for calculating semantic distances between high-dimensional vectors, including Euclidean distance and Cosine distance. Each distance metric is best suited to different application scenarios. Theoretically, cosine similarity is considered the optimal choice for text-based research due to its ability to measure the angle between vectors, which is crucial for understanding semantic relationships. However, clustering algorithms such as hierarchical clustering rely on distance metrics rather than similarity measures. Thus, we transform cosine similarity into semantic distance to ensure compatibility with clustering methods and intuitive interpretation of dendrograms. This approach also aligns with cognitive psychology traditions, where distance-based representations are commonly used to illustrate relationships between constructs ([43]). Therefore, in this study, we define semantic distance as follows ([35]), and a conceptual illustration is provided in Figure 2B(a):Semantic Distance = 1 − cosine similarity(1)

#### 2.2.2. Structure Exploration

A core function of the ESAA is to suggest a framework for categorizing all material items, thereby enabling researchers to gain insights into the underlying data structure. This functionality is achieved through the use of an algorithms, hierarchical clustering.

Hierarchical clustering is an unsupervised clustering method particularly suitable for exploring hierarchical relationships among embedding. Unlike other clustering methods, it does not require pre-specifying the number of clusters, instead building a hierarchical tree (dendrogram) that reveals the multi-level nested structure of data. This study employs a bottom-up (agglomeration) approach, beginning with each data point as its own cluster and progressively merging similar points. This method’s advantage lies in its ability to comprehensively display multi-level relationships among data points, making it especially apt for examining the semantic relationships among psychological scale items.

In hierarchical clustering calculations, various methods are available, including Ward’s method, average method, and complete method. Ward’s method notably excels in minimizing total variance within clusters, resulting in compact, well-defined clusters—a critical aspect for identifying similar items in psychological constructs. Given these merits, we hypothesized that Ward’s method would be the most suitable for the ESAA. Our preliminary experiments confirmed this prediction, demonstrating its superior efficacy. Thus, we have chosen Ward’s method for the ESAA. Further detailed analyses, including outcome metrics and comprehensive methodological frameworks, will substantiate these findings.

### 2.3. Step 3: Output: Synthetic Correlation and Framework

#### 2.3.1. Synthetic Correlation

As a methodological tool for redundancy detection in psychology, the ESAA is designed to report on the semantic relationships between items. This functionality is achieved through the computation of synthetic correlation or semantic distance. Synthetic correlation, a metric introduced in this study, serves as an alternative to traditional correlations derived from empirical data collected from human participants. Synthetic correlation is defined as the cosine similarity between the embeddings. Specifically, Synthetic Pairwise Item Correlation (SPIC) refers to the cosine similarity between a pair of two items (see the conceptual illustration in Figure 2B(b):

Assume that the embeddings for each item *i* and *j* are vectors *v_i_* and *v_j_*, respectively. The formula for calculating the Synthetic Pairwise Item Correlation (SPIC) is:(2)$$SPIC\left(i,j\right)=\cos\left(\theta \right)=\frac{vi\cdot vj}{\left\|vi\right\|\left\|vj\right\|}$$
where *v_i_·v_j_* represents the dot product of *v_i_* and *v_j_*. ‖*v_i_*‖ and ‖*v_j_*‖ are the Euclidean norms (or magnitudes) of *v_i_* and *v_j_*, respectively.

Intra-group (Cluster/Scale) SPIC refers to the synthetic pairwise item correlations calculated between items within the same group, which would be cluster or scale in this study. These correlations are expected to be higher, reflecting the semantic consistency and coherence of items that are theoretically or empirically grouped together.

Inter-group (Cluster/Scale) SPIC refers to the synthetic pairwise item correlations calculated between items from different groups, which would be cluster or scale in this study. These correlations are typically lower, indicating the semantic distinction and divergence between items that belong to different concepts or theoretical categories.

#### 2.3.2. Framework: Solution of Hierarchical Clustering

To analyze the structure of the embeddings, an unsupervised clustering algorithm is needed for pattern mining. Following an extensive review of mathematical theory and performing a series of preliminary experiments, agglomerative hierarchical clustering with Ward’s method was selected. This method constructs a hierarchy of clusters by iteratively merging them based on selected a predefined linkage criterion. The process of this clustering is illustrated in the conceptual diagram shown in Figure 2C. And its calculations were performed using Python 3.11.3 in this study, with the scipy and scikit-learn libraries primarily employed for implementation.

The solution of the hierarchical clustering, which is the major output of the ESAA, referred as the “ESAA-generated Framework (EGF)”, is visually represented as a dendrogram (e.g., Figure 3a). A dendrogram is a tree-like diagram that visually represents the hierarchical relationships among items or clusters, illustrating how individual elements are grouped together based on their similarity or distance.

The term “framework” in this study refers to any classification scheme applied to a given set of items. Theoretically, there can be many frameworks for the same set of items. One particular framework is the outcome of the ESAA, i.e., EGF. On the other hand, a classification scheme where each category corresponds directly to the originating scale of its items is also considered a framework. We refer to this as the “Scale Origin Framework” (SOF). The comparison between EGF and SOF will also be part of the ESAA’s output, as it can reflect potential overlaps between the scales of the original materials.

## 3. Validation of the Proposed Approach

While comprehensive validation through extensive studies is typically necessary for proposing a new approach, such validation is beyond the scope of this initial study. Instead, we focus on the fundamental criteria that an effective method in detecting redundancy in psychological constructs. Specifically, the approach should demonstrate three key capabilities: (1) converging items that are highly correlated, (2) discriminating between items that are uncorrelated, and (3) identifying semantic overlap across different concept that are operationalized as groups of their scale items. To assess these capabilities, we conducted three experiments. The first experiment tested the approach’s ability to converge highly correlated items and to discriminate uncorrelated items. The second experiment evaluated its capacity to identify overlap between items from different scales. Finally, the third experiment served as a robustness check, offering a series of comparative analyses to access of the overall competence of the approach.

The strategy for these tests is theory driven. A robust theoretical foundation regarding correlation, non-correlation, and overlap between concepts provides benchmarks for evaluating the ESAA. If the results produced by the ESAA deviate significantly from theoretical expectations, it indicates that the approach is ineffective. Conversely, if the outcomes align closely with theoretical predictions—showing no evidence to contradict the validity of the ESAA—it supports the approach’s potential for further in-depth investigation.

In prior research on conceptual redundancy, Grit has been widely documented as a relevant construct for such an investigation due to its well-established body of literature and meta-analytic findings. This theoretical grounding ensures that even with a smaller number of items, the results remain interpretable and relevant. Furthermore, expanding the sample size in these preliminary studies would not necessarily provide additional value without compromising the need for clear, theory-based validation. This approach aligns with the perspective of [22] ([22]), who highlight the importance of integrating explanation and prediction in computational social science. By prioritizing the development of an explanatory framework, our work aims to provide meaningful theoretical insights into conceptual redundancy, rather than merely constructing predictive models.

### 3.1. Experiment 1: Convergence and Discriminant Validity

The experiment aims to evaluate the ESAA’s ability to converge highly correlated items and discriminate low-correlated items. If the ESAA has reliable convergence ability, then EGF should categorize those items measuring a same underlying conceptual construct together, with intra-cluster SPIC at least as high as the corresponding intra-scale SPIC. In contrast, if the EGF has sufficient discriminant ability, it should allocate items from scales measuring conceptually distinct constructs into different clusters, with inter-cluster SPIC much lower than corresponding intra-cluster SPIC, and not higher than the corresponding inter-scale SPIC.

#### 3.1.1. Methods

Materials, data, and Appendix A for this study are available through the Open Science Framework: https://osf.io/9fxmq/?view_only=ccb8fc28195c4dadb9ba1df19636591e (accessed on 21 October 2024). Data preprocessing and statistical analyses were conducted using Python (version 3.11.3).

To test the ESAA’s convergence and discrimination abilities, the experimental materials must meet certain criteria: (1) each scale should be widely recognized for high internal consistency, (2) the concepts should exhibit low correlation, as supported by prior studies, and (3) both scales should have an equal number of items.

In line with these criteria, the Conscientiousness and Gratitude scales were chosen:

Conscientiousness Scale: Derived from the NovoPsych Five Factor Personality Scale—30 ([8]), this 6-item scale measures the well-established five factor model of personality (a.k.a. OCEAN). A sample item is “I often do just enough work to get by” (reverse scored).

Gratitude Scale: The Gratitude Questionnaire-Six Item Form (GQ-6: [28]) includes 6 items, with a sample item being “I have so much in life to be thankful for”.

Both scales are well validated with high internal consistency and the two concepts exhibit minimal to no correlation ([1]; [25]), making them ideal for examining convergence and discrimination in psychological research.

Given the nature of these scales, the items measuring Conscientiousness and Gratitude should map distinctly into two separate regions, in the semantic space of embeddings, with the distance between these regions significantly greater than within-region distances.

**Hypothesis** **1.**
*The macro-structure of the EGF will match the SOF; and in the EGF, the inter-cluster SPIC will be significantly lower than the intra-cluster SPIC, with a large effect size.*


#### 3.1.2. Results and Discussion

The hierarchical clustering analysis conducted for these 12 items reveals two distinct clusters, each corresponding precisely to one of the original scales. Embeddings from the same scale are closely grouped together, while embeddings from different scales are clearly separated, as shown in Figure 3a,b. Figure 3a displays the EGF represented as the dendrogram, illustrating the hierarchical clustering of item embeddings using Ward’s method. The embeddings were grouped into two widely separated clusters, corresponded to all items from the Conscientiousness scale and all items from the Gratitude scale, respectively. This clustering pattern aligns with hypothesis.

Figure 3b illustrates the structure of the embeddings in Experiment 1 using an alternative approach: a Kernel Density Estimate (KDE) plot of the two components derived from Principal Component Analysis (PCA). To generate Figure 3b, we followed a three-step process: (1) semantic distance computation, (2) reducing the embedding dimensions to two using PCA, and (3) applying a KDE to the PCA dimensional reduction results, with the color-label for items follows EGF, i.e., the results of previous clustering. The KDE plot was generated using a Gaussian kernel function with an automatically adjusted bandwidth. This plot clearly highlights the separation and cohesion of clusters of the EGF.

As mentioned in the section II of this study, intra-cluster SPIC values represent the semantic coherence within each cluster, while inter-cluster SPIC values reflect the divergence between different clusters. The average SPIC values for the intra-cluster pairs within the Gratitude scale and the Conscientiousness scale (*M* = 0.426, *SD* = 0.0039) were significantly higher than those for inter-cluster pairs (*M* = 0.193, *SD* = 0.0022), demonstrating that item pairs within the same cluster are closer to each other than those between different clusters. To examine whether such difference between intra-cluster and inter-cluster SIC values is significant, we conducted statistic test. Welch’s *t*-test is adopted due to unequal variances and sample sizes. The results revealed a significant difference, *t* (43.35) = 18.46, *p* < 0.001. The effect size, measured by Cohen’s *d*, was 4.50, indicating a very large effect. These findings suggest that intra-cluster similarities are significantly higher than inter-cluster similarities.

These results support H1, demonstrating that the ESSA effectively distinguishes between concepts and converges items within the same concept. Therefore, we can confidently reject the null hypothesis, which suggest that the ESAA lacks convergence or discriminant abilities.

However, it is important to note that these conclusions are limited to the specific materials used in this experiment. Similar to the Turing test, Experiment 1 was designed to explore the capabilities of a new technique. While the results from this single experiment are promising, they only serve as a necessary condition for demonstrating the ESAA’s effectiveness. Extensive empirical validation through future research is required to fully confirm the approach’s generalizability and robustness.

### 3.2. Experiment 2: Overlap-Detection Competence Validation

The aim of this experiment is to assess the effectiveness of the ESAA in detecting semantic proximity between scale items that measure psychological concepts with potential redundancy. These concepts may be so similar that they lack incremental validity—meaning they do not contribute unique information beyond what is already captured by other concepts. As the ESAA is designed to help researchers identify redundancy in psychological concepts, demonstrating this overlap-detection capability is crucial for validating its utility in future studies.

#### 3.2.1. Methods

To assess the ESAA’s capacity for detecting redundancy, it is important to select psychological concepts that have been explicitly recognized in the literature as containing overlapping elements. This ensures a clear benchmark for evaluating the reliability of the ESAA’s results.

The concepts of Grit and Conscientiousness were chosen as they meet this criterion. Grit was doubted for its redundancy with Conscientiousness by many research. A meta-analysis of Grit research ([13]), which analyzed 584 effect sizes across 88 independent samples (totaling 66,807 individuals), revealed that Grit, initially proposed as a higher-order trait predicting of success and performance, shows an excessively strong correlation (*ρ* = 0.84) with Conscientiousness, raising questions about challenging its distinctiveness as a construct.

This experiment utilized the following scales:

Conscientiousness Scale: Same as that in Experiment 1.

Grit Scale: derived from [17] ([17]), this scale contains 12 items divided into two facets: Perseverance of engagement and Consistency of interest, with 6 items per facet. A sample item is “I have achieved a goal that took years of work”.

If there is no overlap between Consciousness and Grit, the data structure should resemble the pattern depicted in Figure 4, where Grit and Conscientiousness are distinct categories, with the two facets of Grit as sub-categories. However, theoretical perspectives suggest that Conscientiousness and Grit may be overlapping or even redundant constructs. As [13] ([13]) noted, the Perseverance facet of Grit and Conscientiousness was reported to have a correlation of *ρ* = 0.89, far exceeds the typical correlation found between two different global measures of Conscientiousness (*ρ* = 0.63) ([30]). Thus, the meta-analysis suggested that “Grit is not a higher-order construct characterized by two lower-order facets” and “may be redundant with Conscientiousness”. Therefore, if the ESAA is capable of identifying redundancy, the EGF should not mirror the SOF in Figure 4 but instead exhibit signs of semantic blending between Conscientiousness and Grit items. This leads to the hypothesis for the current experiment:

**Hypothesis** **2.**
*The structure of the EGF will not be identical to the SOF, with items from Grit and Conscientiousness interfused, rather than separated.*


#### 3.2.2. Results and Discussion

The application of the ESAA on the items from the Grit and Conscientiousness scales resulted in two major clusters, as revealed in the hierarchical structure depicted by Figure 5a,b. The full merging process is visualized in the dendrogram in Figure 5a, which illustrates the hierarchical clustering of item embeddings from the two scales based on Ward’s method. The y-axis represents the dissimilarity at which clusters are merged, while the x-axis shows individual items, color coded for clarity. Items from the Conscientiousness scale are labeled “C”, while those from the Grit scale are labeled “G”. “Cluster 1” in Figure 5 notably includes a mix of items from the Grit scale along with all of the Conscientiousness items. This suggests that, in terms of semantic proximity, some Grit items are closer to Conscientiousness, leading the algorithm to group them together. Specifically, the part of Grit items involving the interfusion are exactly those measuring the so-called Perseverance of Engage facet, which evidence is perfectly consistent with the conclusion of the meta-analysis by [13] ([13]). Figure 5b provides further evidence of semantic overlap, displaying a KDE plot based on the dimension reduction in the embeddings. This figure highlights the interweaving of Grit and Conscientiousness items, confirming the semantic fusion between them. The generation of Figure 5b followed the same three-step process as in Experiment 1, including semantic distance computation, dimensionality reduction, and KDE visualization. These results clearly support Hypothesis 2, which predicted interfusion between items from the Grit and Conscientiousness scales. In sum, the result of the ESAA, i.e., the EGF of Experiment 2, supports H2, being highly consistent with the literature suggesting the redundancy of the Grit concept.

Moreover, EGF performs well in terms of convergence and discriminate validity, as detailed statistics shown in Table 1. The intra-cluster SPIC of EGF (*M* = 0.372, *SD* = 0.132, *n* = 81) was significantly higher than the inter-cluster SPIC (*M* = 0.294, *SD* = 0.071, *n* = 72). Welch’s *t*-Tests were conducted instead of a traditional *t*-test due to the significant heterogeneity of variance observed in the data, *t* (126) = 4.572, *p* < 0.001. The large effect size (Cohen’s *d* = −0.717) indicate that the pairwise semantic similarity of items within and between cluster in EGF differ significantly. Further, the difference between intra-scale and inter-scale SPICs for SOF also reached significance, but the all the statistic values indicating performance of framework are inferior to those of EGF.

In summary, the results of Experiment 2 demonstrate that the ESAA successfully identified semantic interfusion between the Grit and Conscientiousness scales, as evidenced by the hierarchical structure and reduced dimensional KDE plots. Hypothesis 2, which predicted this interfusion, was supported by the formation of the mixed-source cluster, highlighting the overlapping semantic nature of the two concepts. Furthermore, statistics of SPICs demonstrate the validity of this EGF. Overall, the results in Experiment 2 indicate that the ESAA has potential to serve as a reliable tool for detecting redundancy among psychological constructs, addressing this study’s objective of validating the ESAA’s overlap-detection competence.

### 3.3. Experiment 3. Robust Check and Comparative Analysis

The results from the previous two experiments present the ESAA as a potential tool for redundancy research on psychological concepts. However, before drawing a validating conclusion, two arguments remain: (1) Are the results from the first two experiments robust? Experiments 1 and 2 selected corresponding experimental materials independently. However, if the experimental materials were input in a different way, will the ESAA’s calculations remain stable? This is the “robust” argument. (2) Does the ESAA provide added value compared to baseline tools? A sample and intuitive baseline is the use of chatbots based on LLMs, such as ChatGPT. If similarly effective analytical results can be generated through simple prompts, there may be no substantial benefit in developing a new tool. Therefore, it becomes crucial to compare the performance of the ESAA with existing chatbot. The comparison will help determine whether the ESAA offers genuine innovations and improvements. This is the “incremental value” argument.

To address these questions, we designed a series of trials in this experiment applying the ESAA and alternative approaches on same input material, and comparing the performance of their outputs, namely the EGF and other frameworks. The hypotheses are that the ESAA has robustness and incremental value, which means that the outputs of the ESAA are consistent with each other and the performance of EGF beats any other frameworks in terms of convergence, discriminate, and redundancy detection performance.

#### 3.3.1. Material and Procedure

The materials for this experiment are a combination of those used in the previous two experiments, incorporating three scales measuring Conscientiousness, Grit, and Gratitude (24 items in total). The procedure involved generating the frameworks, followed by three series of trials comparing the EGF with the frameworks generated by alternative approaches.

The robustness of the ESAA was evaluated by comparing the EGF in from Experiments 3 from those from Experiments 1 and 2. We expected the EGF of the current experiment to align with the previous ones, confirming the stability of the ESAA’s output. Secondly, to assess the ESAA’s incremental value, we generated two baseline frameworks using the most advanced LLM-based chatbots (GPT-4.0) in present, i.e., ChatGPT-4o and -o1, anticipating that the chatbot-generated frameworks would be inferior to EGF in terms of the convergence, discrimination, and interpretability.

#### 3.3.2. Results and Discussion

The clustering results in Figure 6a,b clearly reveal the hierarchical structure of the EGF for Experiment 3. Figure 6a presents the dendrogram of the ESAA analysis, which illustrates the hierarchical clustering of item embeddings from the Conscientiousness and Gratitude scales, based on Ward’s method. The y-axis shows the semantic distance between clusters, while the x-axis lists individual item embeddings, color coded by clusters. Items from the Conscientiousness scale are labeled “C”, Gratitude scale items are labeled “G”, and the two facets of Grit—Consistency of Interest and Perseverance of Effort—are labeled with “I” and “E”, respectively. When the clustering threshold was set at 0.7 for the mean distance between clusters at merge, the items were divided into four distinct groups, corresponding to the original subscales. However, a closer look at the dendrogram reveals some semantic blending, particularly between Grit and Conscientiousness items. For example, the Perseverance of Effort facet of Grit initially clusters with Conscientiousness items before later merging with the Consistency of Interest facet, indicating a notable overlap between these two constructs. This merged group is then combined with the Gratitude items at a more distant clustering level.

The step-by-step merging process is further visualized in Figure 7a–c, which depict the distribution of data points after reducing the high-dimensional embedding space to two dimensions using PCA. In Figure 7a, the data points are divided into four classes, matching the original subscales. As seen in Figure 7b, when the clustering is reduced to three classes, the Conscientiousness items merge with the Perseverance of Effort facet from the Grit scale, reflecting their semantic proximity. Finally, Figure 7c illustrates the division into two major clusters, where the combined Grit and Conscientiousness items form a single category that merges with Gratitude at a higher level of abstraction. This merging pattern highlights the structural relationships between the concepts, suggesting significant overlap between Grit and Conscientiousness, while the separation from Gratitude demonstrates the ESAA’s ability to discriminate between more distinct concepts. This result showcases the ESAA’s effectiveness in capturing both convergence and distinction across psychological scales, particularly in identifying concepts with overlapping semantic features.

The information suggested by EGF for overlap shows statistical significance. Specifically, the facet E items from the Grit scale first merges with the Conscientiousness scale items, resulting in the formation of cluster 2.2. This cluster 2.2 (*M* = 0.535, *SD* = 0.090) exhibits a higher mean of intra-group SPICs compared to the Grit scale (*M* = 0.355, *SD* = 0.138), with a significant difference observed (Cohen’s *d* = 1.382, *p* < 0.001).

Additionally, the discriminant validity among the (sub)clusters was also confirmed. The intra-group Synthetic Pairwise Item Correlations (SPICs) for each (sub)cluster were aggregated and used as a baseline for comparison, and the results indicated significant discriminant validity for all comparisons (*p* < 0.001). The effect sizes for these comparisons, as shown in Table 2, were substantial, with Cohen’s d values indicating large effects across all clusters.

The comparative analysis results were fully in line with expectations and are further summarized in Table 3, which highlights the consistency of various framework outputs with theoretical expectations. The robustness check confirmed that the EGF from Experiment 3 was entirely consistent with those from Experiments 1 and 2, with Gratitude items forming a distinct sub-cluster, identical to the EGF structure from Experiment 1. This consistency reinforces the stability of the ESAA’s calculations.

In terms of incremental value, the ChatGPT models (versions 4o and o1 preview) failed to generate frameworks that aligned with theoretical expectations as effectively as the ESAA, confirming the ESAA’s superiority in terms of convergence, discrimination, and interpretability. Although the ChatGPT-generated frameworks captured the macro-structure, they were unable to provide consistent sub-structures, thus falling short of the precision achieved by the ESAA.

Detailed outcome frameworks generated by all the alternative approaches can be seen in the Appendix A at https://osf.io/9fxmq/?view_only=ecef150ef2184d0da5106b6413f093c4 (accessed on 21 October 2024).

## 4. Discussion

The primary aim of this study is to propose and evaluate an approach for detecting redundancy in psychological concepts by utilizing the latest advancements in AI technology development. The proposed method, referred to as the Embedding-based Semantic Analysis Approach (ESAA) is a multi-step process. Initially, the ESAA takes as input the textual content of psychological scale items and transforms them into high-dimensional vectors by leveraging the GPT-3 large language model. This transformation effectively converts qualitative textual data into computable numerical representations that capture their semantic meanings. Subsequently, the ESAA applies unsupervised clustering algorithms—particularly hierarchical clustering—to these embeddings to uncover latent semantic structures among the items. Finally, the output includes an analysis of the semantic relationships among these items, encapsulated in a framework referred to as EGF, as well as synthetic correlations.

Through a series of three experiments, preliminary evidence attests to the ESAA’s usefulness and reliability. The ESAA successfully converged semantically similar items, discriminated between items with significant semantic differences, and identified patterns of semantic overlap among constructs known to have theoretical redundancies. These findings suggest that the ESAA has the potential to serve as a tool for researchers aiming to refine theories and reduce redundancies in psychological measurement.

Some features of the ESAA distinguish it from contemporary research, which, to the best of our knowledge, also exploits embeddings. The first one is the ESAA’s design philosophy: simplifying the processes of vectorization, clustering, and evaluation to ensure accessibility for psychological researchers without requiring extensive computational expertise. Unlike the methods that rely heavily on fine-tuning language models like BERT or LLM ([23]; [42]), the ESAA uses the latest API-based language models, which continuously evolve with advancements in the field. This approach eliminates the need for resource-intensive retraining and empowers psychological researchers by providing a practical and streamlined manual to guide redundancy detection. Our intention is to keep the core work of redundancy detection within the domain of psychological researchers, allowing them to leverage automated tools while retaining control and interpretability of the process. The ESAA is designed as an aid, not a replacement, supporting psychologists in analyzing item content in a more automated and repeatable way, as opposed to the time-intensive manual efforts traditionally required. By combining cutting-edge NLP techniques with an accessible and operational framework, the ESAA bridges the gap between advanced computational tools and psychological research, ensuring the redundancy detection process remains grounded in the expertise of domain professionals.

Secondly, while contemporaneous studies (e.g., [42]) rely on large-scale empirical data from public databases, we have deliberately chosen to evaluate the reliability of the ESAA outcomes through psychological theory-based predictions. This choice underscores the importance of the researcher’s role in the testing process. Psychological researchers bring theoretical knowledge and critical judgment, which adds a unique and valuable perspective to redundancy detection. By incorporating this perspective, the ESAA provides a meaningful complement to purely data-driven evaluations, offering insights that are both theoretically grounded and practically applicable.

To sum up, the ESAA exhibits the potential to contribute to the development of psychological theoretical research, particularly areas such as conceptual redundancy reduction and integration studies. This approach offers unique value through its ability to perform analyses based solely on the content of the psychological scales, thereby eliminating the need for traditional data collection from participants in psychological research. This would significantly reduce research costs. Furthermore, by mitigating subjective bias, ESSA would enhance the reproducibility of findings and improves the reliability of research conclusions.

The ESAA also represents a meaningful step forward in psychological measurement by addressing redundancy and refining the core components of scales, it provides new opportunities for optimizing psychological instruments and improving data quality. Scales are commonly used data collection tools in research across psychology, sociology, education, and other fields, where studies often require the collection of multi-faceted information. However, the overlap among scales measuring various aspects can lead to substantial cost inefficiencies in terms of participants’ engagement. The ESAA can facilitate the design of large-scale streamlined measurement instruments for these studies at negligible cost, resulting in a substantial reduction in the number of items in the processed questionnaire format while preserving the richness of the information obtained.

However, the current research remains in the initial stage and suffers considerable limitations in the following aspects. First, before a definitive judgment regarding the ESAA’s reliability can be made, further validation through comprehensive testing on diverse datasets and psychological constructs is needed. This study uses psychological theoretical predictions as the standard for assessing the ESAA’s performance, rather than making direct comparisons with traditional methods. While this approach offers clear advantages for an initial investigation, it lacks generalizability. Moreover, the reliance of this study on specific psychological scales raises questions about the generalizability of the findings to other constructs. Furthermore, the generalizability of the findings in this study to disparate languages or cultural contexts remains uncertain. The current study focuses solely on English-language scales, and linguistic structures, cultural-specific concepts, and expressions can vary significantly across different languages and cultures. Consequently, the ESAA’s applicability and effectiveness in non-English or culturally diverse contexts have yet to be tested. Additionally, it is noteworthy that future research employing the ESAA with proprietary scales, such as the MBI or BDI-II, must ensure compliance with copyright regulations and obtain the necessary permissions before incorporating such materials into their analyses.

Secondly, the ESAA lacks control and optimization over the quality of embeddings, which entails certain risks. The ESAA directly uses embeddings generated by the GPT-3 Large model; however, any large language models (LLMs) may introduce biases in language representation, potentially affecting the semantic analysis outcomes. These biases could stem from the training data and might influence the model’s understanding of certain terms, leading to skewed findings. Additionally, in this study, the ESAA generated embeddings using LLM only once, which may introduce sample bias. Technically, if the order of the input text is disrupted, the resulting embeddings are likely to differ because embeddings are generated based on the content of the text and its context; thus, the sequence of the text affects its semantic representation. Disrupting the order may lead to variations in the model’s understanding of the text, resulting in the generation of different embedding vectors. Although such perturbations may not significantly impact the semantic relationships between embeddings, a systematic evaluation of the specific effects of these perturbations must be conducted before the ESAA can be claimed as a reliable tool. This aspect has not been addressed in the current study and must be considered in future research.

Third, the impact of text preprocessing steps on the results was not thoroughly examined. Different preprocessing techniques, such as lemmatization, removal of stop words, could influence the embeddings generated by the LLMs, potentially affecting the semantic analysis outcomes. Future studies should systematically investigate how various text preprocessing methods affect the ESAA’s performance.

Future research involves several tasks that need to be addressed. Validation, application, and iterative improvement are all important directions requiring effort, and the work aimed at these goals may overlap. Among these, some constitute fundamental and essential research. For example, it is crucial to incorporate empirical participant data, such as item response patterns and inter-item correlations, to cross-validate the ESAA’s outputs. Second, we propose applying the ESAA to more complex scales with less overlap and more nuanced constructs to evaluate its performance in diverse and challenging contexts. For instance, longer scales or item pools like PROMIS (Patient-Reported Outcomes Measurement Information System), designed with numerous items across health domains, pose challenges for manual redundancy detection due to their complexity and volume. AI-based tools like the ESAA have the potential to offer a scalable and objective solution for identifying redundancy and refining constructs, making them particularly promising for large-scale psychometric evaluations. Third, we aim to extend the ESAA to cross-linguistic testing by analyzing psychological scales across different languages. Collectively, these efforts will contribute to refining the ESAA as a reliable and versatile tool for redundancy detection and theoretical advancement in psychological research.

Aside from research focusing on tool evaluation, another research opportunity arises to utilize and upgrade the ESAA to conduct meaningful research addressing theoretical and practical problems in psychology. As previously mentioned, our research aims to serve psychological researchers by enabling them to harness the power of AI while retaining control and interpretability of the research process. With attainable technical upgrades to the ESAA, it could soon facilitate numerous psychological studies. For example, a considerable number of studies rely on questionnaires that need to collect empirical data on multiple psychological constructs simultaneously. However, the feasibility of these studies is greatly limited by the length of questionnaires that human participants can tolerate. The ESAA’s capabilities on detecting extremely similar items across scales for different constructs may make a previously impossible task achievable: obtaining data equivalent to that from full-length questionnaires through significantly reduced number of questionnaire items. We call for cooperation on this type of research.

The ultimate aspiration that motivates the creation of the ESAA—namely, the redundancy reduction in psychological concepts and measurements, as well as the refinement of theory—holds immeasurable value for the advancement of psychology. The proposal and validation of a tool for redundancy detection represent merely the first small step toward this vision. A substantial amount of research work remains to be accomplished, and it is hoped that scholars will join in this endeavor.

## Figures and Tables

**Figure 1 jintelligence-13-00011-f001:**
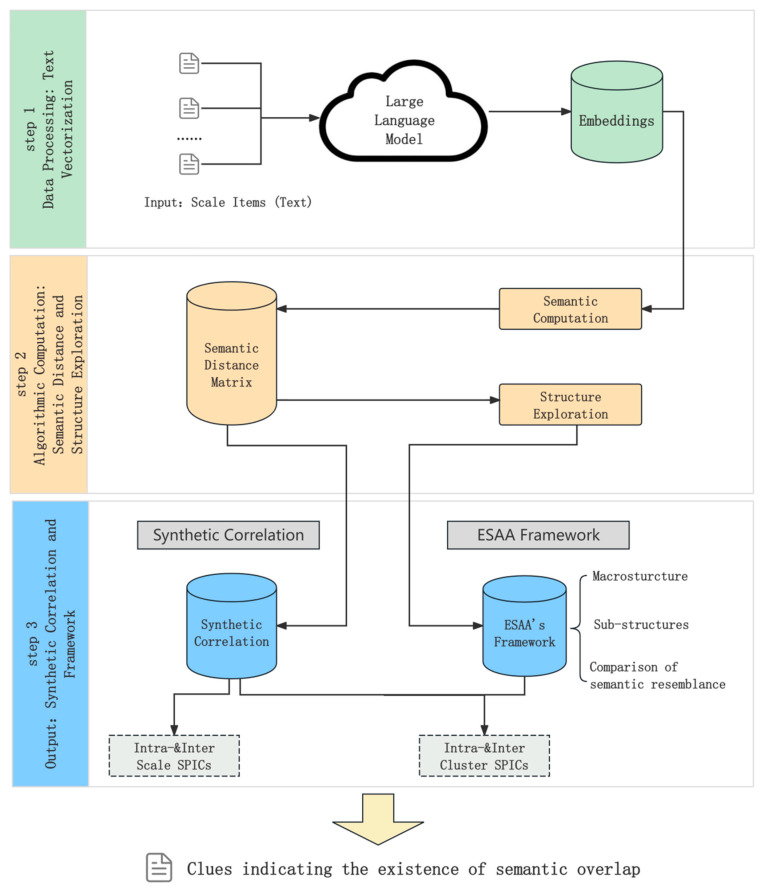
Workflow of the Embedding-based Semantic Analysis Approach (ESAA) for Redundancy Detection in Psychological Scales.

**Figure 2 jintelligence-13-00011-f002:**
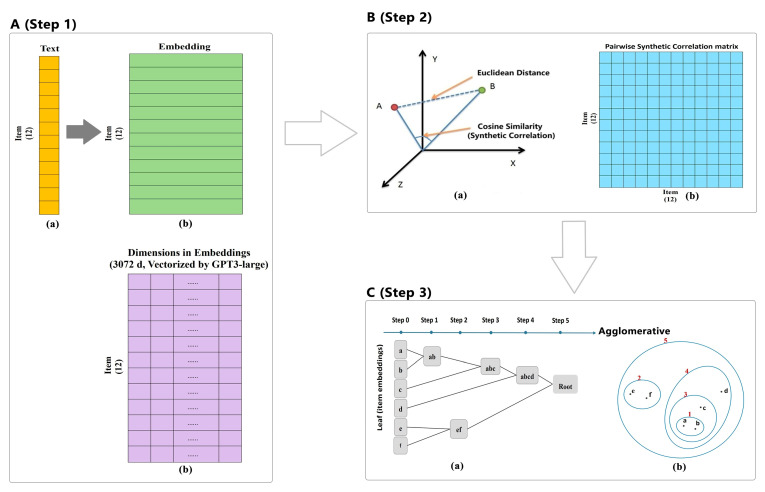
Conceptual Diagram for the ESAA Using Material of Experiment 1 as an Example. Panel (**A**) (Step 1): (a) Input text (12 items) is transformed into embeddings; (b) embeddings are represented as high-dimensional vectors. The green plot provides a general visualization, while the purple plot adds internal details (embeddings generated by GPT-3-large). Both plots depict the same 12 embeddings but highlight different aspects. Panel (**B**) (Step 2): (a) Semantic distances (Euclidean distance and cosine similarity) are visualized conceptually; Colored points represent individual embeddings (high-dimensional vectors) projected into a three-dimensional space for illustration purposes. (b) Pairwise synthetic correlation matrix is generated. Panel (**C**) (Step 3): Process and method for the hierarchical clustering. The points (e.g., a, b, c) represent individual embeddings. (a) Agglomerative hierarchical clustering is demonstrated step-by-step; (b) clustering process visualized as hierarchical merging.

**Figure 3 jintelligence-13-00011-f003:**
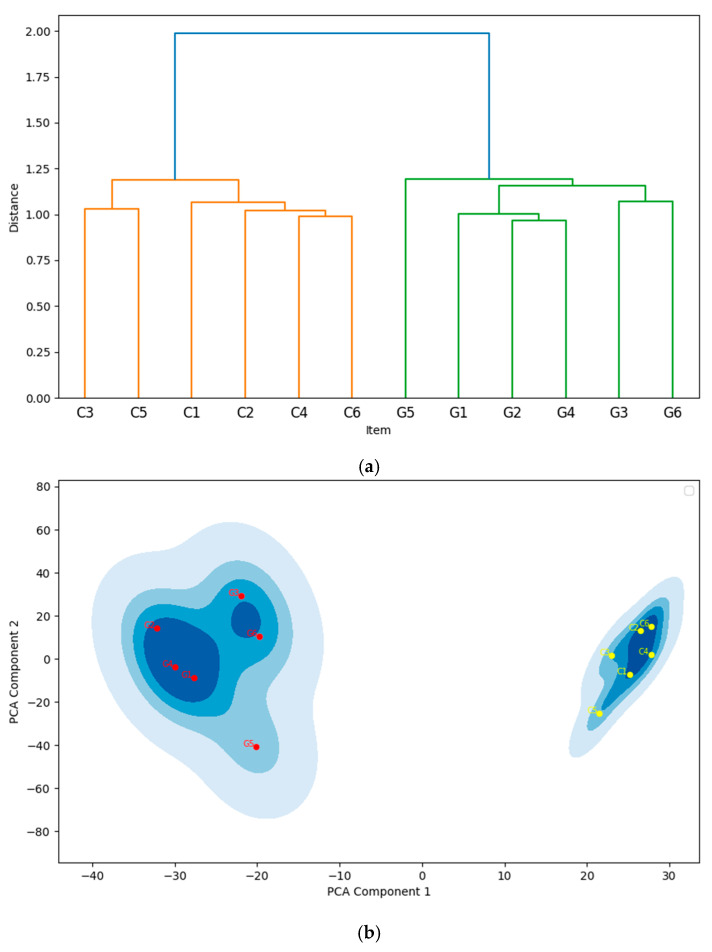
(**a**) The ESAA-Generated Framework in Experiment 1. This figure shows the dendrogram resulting from the hierarchical clustering of the item embeddings from the Conscientiousness and Gratitude scales, using Ward’s method. The horizontal axis represents the items, labeled with a “C” prefix for items from the Conscientiousness scale and a “G” prefix for items from the Gratitude scale. Details of the labels and their corresponding item contents can be found in Appendix A. The vertical axis represents the semantic distance between the item embeddings at merge. The clusters are color coded for clarity. (**b**) KDE plot of PCA Components in Experiment 1. This figure displays kernel density estimation (KDE) contours of item embeddings projected onto the first two principal components. Data points representing the item embeddings are colored according to their assigned clusters from EGF. Varying shades of blue represent the density of item embeddings in the area, with deeper blues indicating regions with more nearby items, while lighter blues signifying areas of lower data density.

**Figure 4 jintelligence-13-00011-f004:**
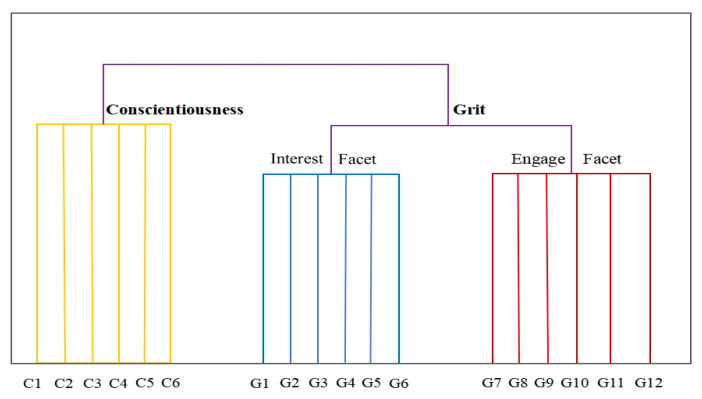
The Scale Origin Framework for Experiment 2. This figure illustrates the structural pattern if there is no overlap between Consciousness and Grit.

**Figure 5 jintelligence-13-00011-f005:**
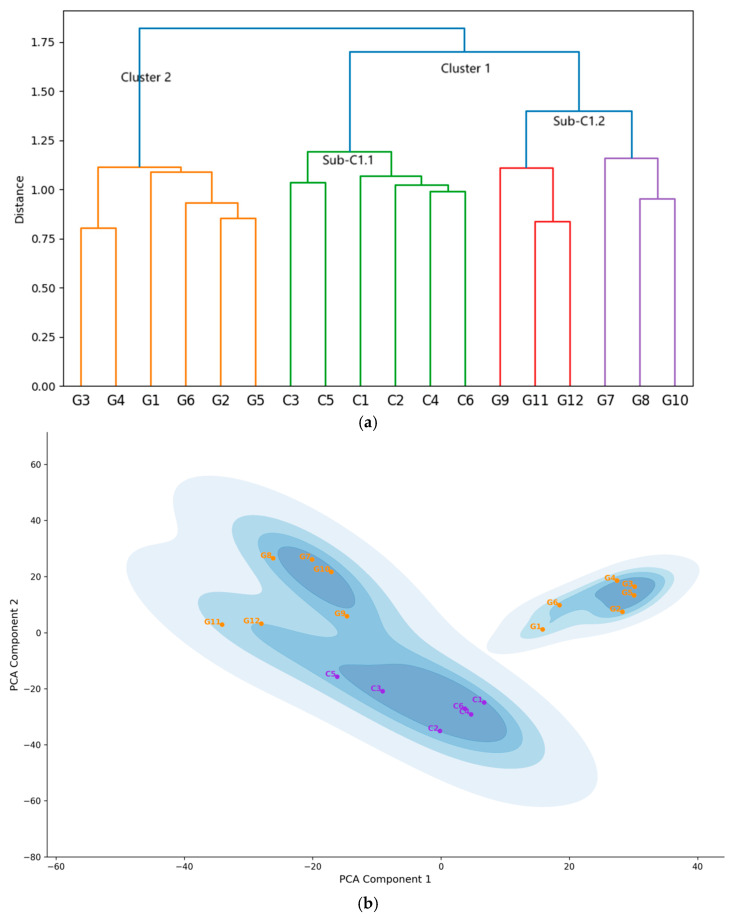
(**a**) The ESAA-Generated Framework in Experiment 2. This figure shows the dendrogram resulting from the hierarchical clustering of the item embeddings from the Conscientiousness and Grit scales, using Ward’s method. The horizontal axis represents the items, labeled with a “C” prefix for items from the Conscientiousness scale and a “G” prefix for items from the Grit scale. Details of the labels and their corresponding item contents can be found in Appendix A. The vertical axis represents the semantic distance between the item embeddings at merge. The clusters are color coded for clarity. (**b**) KDE plot of PCA Components in Experiment 2. This figure displays kernel density estimation (KDE) contours of item embeddings projected onto the first two principal components. Data points representing the item embeddings are colored according to their assigned clusters from hierarchical clustering. Varying shades of blue represent the density of item embeddings in the area, with deeper blues indicating regions with more nearby items, while lighter blues signifying areas of lower data density.

**Figure 6 jintelligence-13-00011-f006:**
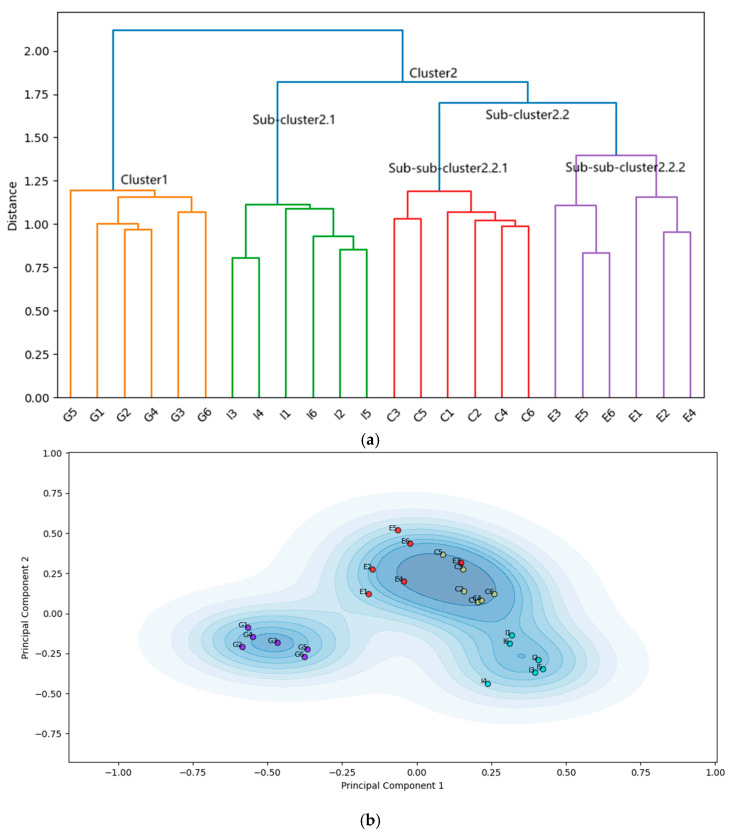
(**a**) The ESAA-Generated Framework in Experiment 3. This figure shows the dendrogram resulting from the hierarchical clustering of the item embeddings from the Gratitude, Conscientiousness and Grit scales, using Ward’s method. The horizontal axis represents the items, labeled with a “C” prefix for items from the Conscientiousness scale, a “G” prefix for items from the Gratitude scale, and “I” and “E” prefix representatively for items from the two facets of Grit scale: persistency of interest and engagement. Details of the labels and their corresponding item contents can be found in Appendix A. The vertical axis represents the semantic distance between the item embeddings at merge. The clusters are color coded for clarity. (**b**) KDE plot of PCA Components in Experiment 3. This figure displays kernel density estimation (KDE) contours of item embeddings projected onto the first two principal components. Data points representing the item embeddings are colored according to their assigned clusters from hierarchical clustering. Varying shades of blue represent the density of item embeddings in the area, with deeper blues indicating regions with more nearby items, while lighter blues signifying areas of lower data density.

**Figure 7 jintelligence-13-00011-f007:**
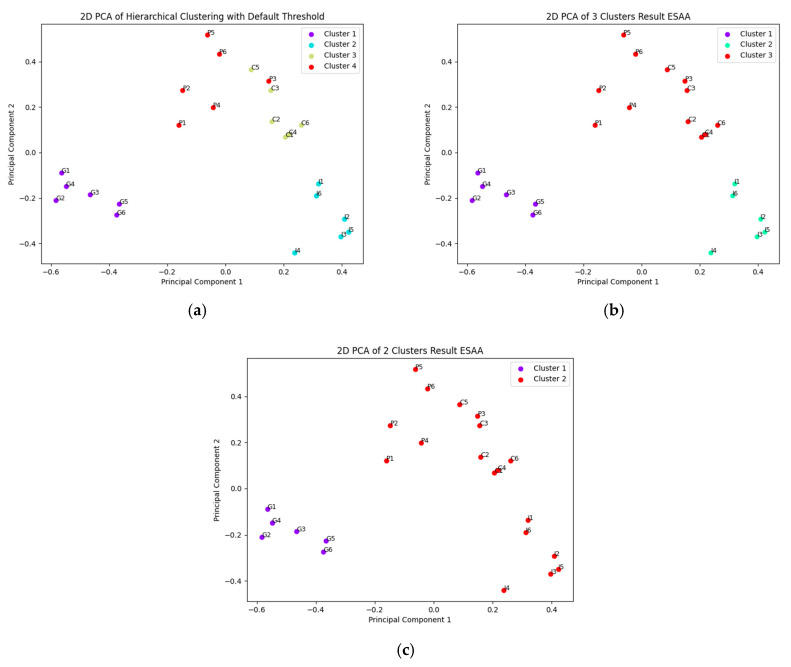
Stepwise Clustering and Dimensional Reduction in Item Embeddings in Experiment 3. (**a**) Clustering into four classes corresponding to the original subscales; (**b**) clustering into three classes, showing the merging of Conscientiousness with the Perseverance of Effort facet from Grit; and (**c**) final clustering into two major categories, highlighting the integration of Grit and Conscientiousness with Gratitude.

**Table 1 jintelligence-13-00011-t001:** SPICs Comparison between EGF and SOF.

Framework	SPIC	Mean	SD	n	t(df)	*p*-Value	Cohen’s d
EGF	Intra-cluster	0.372	0.132	81	--	--	--
	Inter-cluster	0.294	0.071	72	--	--	--
	Difference (1)	0.077			4.572(126)	0.000	0.717
SOF	Intra-scale	0.370	0.130	81	--	--	--
	Inter-scale	0.297	0.078	72	--	--	--
	Difference (2)	0.073			4.284(134)	0.0000	0.675

**Table 2 jintelligence-13-00011-t002:** Statistics on the Key SPICs of the EGF in Experiment 3.

Inter-Group SPIC	Mean	SD	Difference with Baseline	Cohen’s d	*p*-Value
Cluster1 to Cluster2	0.193	0.050	0.251	−3.400	0.000
Cluster 2.1 to Cluster 2.2	0.294	0.071	0.150	−1.744	0.000
Cluster 2.2.1 to Cluster 2.2.2	0.270	0.080	0.174	−1.878	0.000

**Table 3 jintelligence-13-00011-t003:** Comparative Analysis of Various Framework Results with Respect to Their Consistency with Theoretical Expectation.

Output of the Approach	Macro-Structure	Sub-Structures
EGF of Exp3	√	√
EGF of Exp1 or Exp2	√	√
Framework by ChatGPT-4o	√	×
Framework by ChatGPT-o1 preview	√	×

Note. In the “Output of the Approach” column of the above table, unless otherwise specified, the term “approach” refers to that used in Experiment 3. √ indicates consistency with theoretical expectations; × indicates inconsistency with theoretical expectations.

## Data Availability

The data generated and analyzed in this study are partially available within the Appendix A. For additional data supporting the findings of this research, interested parties may contact the corresponding author, who will provide the information upon receiving a reasonable request.

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
