# Peer review of "An Embedding-Based Semantic Analysis Approach: A Preliminary Study on Redundancy Detection in Psychological Concepts Operationalized by Scales"

_jintelligence, 2025, doi:10.3390/jintelligence13010011_

Round 1
Reviewer 1 Report
Comments and Suggestions for Authors
The manuscript fails to adequately contextualize ESAA within the broader field of redundancy detection methodologies in psychology. Although the manuscript acknowledges the limitations of traditional methods, such as factor analysis and expert judgment, it lacks a comprehensive examination of the theoretical justifications for why an embedding-based approach is uniquely advantageous. A further comparison with other contemporary computational methods in psychology would serve to reinforce the argument for ESAA.
The selection of GPT-3 as the primary embedding model is acknowledged, yet the rationale behind this decision is insufficiently explained. There is a paucity of discussion regarding the rationale behind the selection of this model over other, potentially more domain-specific or open-source alternatives, such as BERT or RoBERTa. This omission allows for the possibility of questioning the generalizability and appropriateness of the selected model for psychological scales in particular.
The manuscript presents the results as being highly conclusive, despite the fact that the experiments are still in their preliminary stages. The assertions regarding ESAA's efficacy should be more circumspect, with a more explicit recognition that more comprehensive testing is necessary to substantiate its utility beyond the initial experiments.
While figures such as dendrograms and KDE plots are employed, the interpretations thereof are not sufficiently elucidated. The provision of supplementary labeling, annotations, or step-by-step descriptions of these visuals would facilitate comprehension, particularly for readers less familiar with embedding-based methods.
The manuscript asserts that ESAA offers substantial incremental value; however, it lacks sufficient concrete evidence to substantiate this claim. To reinforce this argument, it would be beneficial to present specific instances where ESAA has demonstrated superior performance in redundancy detection compared to existing, more straightforward tools.
While the manuscript does mention some limitations, a more candid and detailed examination of the potential shortcomings of ESAA is warranted. For instance, the impact of text preprocessing, potential biases in the GPT-3 model, and the generalizability of findings to disparate languages or cultural contexts warrant more comprehensive examination.
While the manuscript does acknowledge the necessity for broader validation, it lacks a comprehensive roadmap outlining future research directions. It would be beneficial to suggest specific next steps for the authors to consider, such as using ESAA on more complex or less overlapping scales, incorporating empirical participant data, or cross-linguistic testing. Doing so would enhance the quality of this section.
The cross-validation of ESAA outputs with participant-derived correlation data could facilitate the integration of theoretical and practical perspectives, thereby enhancing the appeal of the approach to practitioners in psychological research.
Reviewer 2 Report
Comments and Suggestions for Authors
I like this article quite a bit in terms of its originality. Being able to understand item content in a more automated and repeatable way than just having grad students poring through the content over a conference room table would be extremely helpful. Using LLMs to aid researchers (rather than trying to replace them) is a responsible way to go and I want to encourage the authors to elaborate this point.
I think that my main reservations about it are thoroughness of documentation of what is being done as well as the artificially small nature of the examples.
This article would benefit substantially from providing more information to the reader about how an LLM works, how clustering is done, as well as what output is actually being clustered. This is not particularly well-documented or explained. I get why Ward's method was chosen but readers not familiar with clustering would not from the rather casual discussion and lack of references. Also, Ward's method optimizes the criterion that maximizes between-group variance. It doesn't somehow favor that---it was literally built to do that! How was the clustering done, e.g., in R using the cluster package?
Beyond that, the examples are pretty small scale. I appreciate that for the first example or two, but where a procedure like this would truly shine would be in dealing with things like long clinical scales that are notorious for having many redundant items. A bigger scale example would be extremely important in my view.
I do think that LLMs are notorious for having copyrighted material needs to be addressed. Lots of legacy scales like the MBI or BDI2 are copyrighted. This is a challenge that potential users need to face.
Finally, I think better connecting to other AI related literature in psychology and education would be helpful. As an example, Matthias von Davier (current head of TIMSS/PERLS) has done a lot of experiments using earlier versions of generative AI to create items; in TIMSS these are used as prototypes and subjected to further field testing. It would be useful to examine how LLMs can be used in scale analysis and scale construction.
Reviewer 3 Report
Comments and Suggestions for Authors
Thanks for the opportunity to review this work. The manuscript uses Large Language Models ("LLMs", specifically GPT3) to get semantic vectors representing the items in psychological scales. Then, the authors use hierarchical clustering to evaluate the degree to which items overlap (and don't) as a step towards identifying redundancy in item pools.
My qualifications for evaluating this work are based on my familiarity with traditional methods of evaluating structure (and it's many forms) among psychological assessment content, and based on my exclusive focus on the use of LLMs in psychological assessment over the last 3 years. In fact, I have actually done the some of the analyses I read about in this manuscript.
While I'm generally supportive of the ideas underlying this work, I have 3 "sets" of concerns. The first is that this manuscript seems poorly suited for this outlet. I realize that there is some leeway, but it truly doesn't fit in terms of content in my opinion.
The second is that I'm aware of a registered report (now in Stage 2) that has already reported on extremely similar analyses. To be clear, I'm not an author on this other work. But, it's very substantially overlapping and it has been discussed quite a lot publicly among psychological assessment researchers. I don't know what to recommend in this case exactly. There is typically room for multiple research groups to publish on the same concepts, but the current work leans heavily on the idea that their contribution (ESAA) is highly novel. Given that the other work is cited in this manuscript, I find the claim of novelty unconvincing.
The last set of concerns is more standard -- just the typical kind of reviewer feedback.
1. I'd like to hear more about how the authors are vectorizing the items. As I mentioned, I have done this work myself, mainly for a paper published a few years ago using personality adjectives (I won't go so far as to request that the authors cite it but I do think this paper is missing a more robust lit review...). A challenge that came up repeatedly when working with phrased items (like the ones used here) is that gpt3 did not handle negation well. It often ignores it so items that should be negatively correlated (or using their methods... semantically distant) were actually highly correlated. This is a pretty well known problem with gpt (and this is the reason many teams have switched to models trained specifically to handle content like this). How did the authors get around negation?
2. What is the logic for using 1-cosine similarity instead of just cosine similarity?
3. Why not use more than 12 items in all the studies? It's free, does not require an IRB, and takes little time - I'd like to see a more meaningful contribution to the literature than such a trivial test case.
Round 2
Reviewer 1 Report
Comments and Suggestions for Authors
The authors have addressed the comments, and the manuscript has been sufficiently improved for its publication. I appreciate the authors efforts on this manuscript.
Author Response
The authors have addressed the comments, and the manuscript has been sufficiently improved for its publication. I appreciate the authors efforts on this manuscript.
Response: We sincerely thank you for your feedback and support.
Reviewer 2 Report
Comments and Suggestions for Authors
The authors have largely satisfied my concerns.
I do think it would be worth mentioning the challenges of going to a longer scale, which is where a tool like AI can be extremely helpful compared to existing practice. In a very short scale like Grit, it's not too hard to detect redundancy by eye, though I guess the GPT-based approach does have the benefit of doing it in a more objective manner. Larger problems occur with many items in, say, scale development, where researchers are trying to cull or rephrase, or in an item pool situation like PROMIS, where there are many items by design.
I feel confident the editors can handle any remaining issues.
Author Response
The authors have largely satisfied my concerns.
I do think it would be worth mentioning the challenges of going to a longer scale, which is where a tool like AI can be extremely helpful compared to existing practice. In a very short scale like Grit, it's not too hard to detect redundancy by eye, though I guess the GPT-based approach does have the benefit of doing it in a more objective manner. Larger problems occur with many items in, say, scale development, where researchers are trying to cull or rephrase, or in an item pool situation like PROMIS, where there are many items by design.
I feel confident the editors can handle any remaining issues.
Response: Thank you for your feedback. We have added a discussion on the potential future work of applying AI to longer scales and item pools to acknowledge your insightful points (see lines 771-776).